# The acceptability of intimate partner violence screening and response among refugee women accessing Australian resettlement services

**Nigel Spence**[1]*, Jo Spangaro[1], Nicola Man[2], Jacqui Cameron[1], Kelsey Hegarty[3], Jane Koziol-McLain[4], Tadgh McMahon[5], Astrid Perry-Indermaur[6], Chye Toole-Anstey[1], Jeannette Walsh[1], Anthony Zwi[7]

1 School of Health and Society, University of Wollongong, NSW, Australia, 2 National Drug and Alcohol Research Centre, University of New South Wales, NSW, Australia, 3 Division of Medicine and Health Sciences, University of Melbourne, Victoria, Australia, 4 Centre for Interdisciplinary Trauma Research, Auckland University of Technology, Auckland, New Zealand, 5 Department of Research Strategy and Policy, SSI, NSW, Australia, 6 Division of Women, Equity and Domestic Violence, SSI, NSW, Australia, 7 School of Social Sciences, University of New South Wales, NSW, Australia

* nspence@uow.edu.au

**Data Availability Statement:** All relevant data are within the manuscript.

## Abstract

Screening and response for intimate partner violence (IPV) is recommended for women in priority populations and is implemented in health services across diverse jurisdictions. Most women experiencing IPV strongly support screening, however this is untested with refugee women in resettlement contexts. Around one third of refugee women in Australia experience IPV and face multiple post-migration challenges. SAHAR (Safety and Health after Arrival) tested IPV screening using the ACTS tool, brief intervention, and referral with women accessing four settlement support services. Women attending sites during the study period were invited to participate in a three month follow up survey with participation by 321/375 women recruited (86%). Acceptability of IPV screening was assessed against (i) levels of comfort with the IPV screening questions and (ii) strength of agreement/disagreement with settlement services asking women about being frightened, controlled or hurt by their partners. Of participants who recalled the screening, 93% reported being very or moderately comfortable with being asked the questions (89% who had experienced IPV; 94% of those with no IPV identified). For all participants, 84% strongly or somewhat agreed with services asking the questions, with no significant difference in agreement between disclosing and non-disclosing groups. Those with no recall of the questions had lower overall agreement and higher disagreement than those who recalled the questions (88% and 10%). Acceptability was not associated with age, country of origin, household composition, time since arrival or number of prior service visits. Participants identified 'care shown by the worker', 'talking to someone in my own language', 'trust in the privacy of the service', and 'talking to a female worker' as the factors most important for encouraging discussion of IPV. High acceptance of IPV screening by refugee women supports consideration of implementation across settlement services, a key access point for refugee women with diverse needs.

**Funding:** This study was funded by the Australian Research Council (LP190101183) and SSI, a large non-government agency providing settlement services based in Australia. The Australian Research Council had no role in study design, data collection and analysis, decision to publish, or preparation of the manuscript. SSI had two chief investigators on the study governance committee who contributed ideas to the study's implementation, data analysis and preparation of this manuscript, while at all times the University of Wollongong investigators retained full control of study design, implementation, data collection and analysis, decision to publish, and preparation of the manuscript.

**Competing interests:** The authors have declared that no competing interests exist.

# Introduction

Physical or sexual violence against women by a current or former partner is experienced by 30% of women globally, leading to severe physical and mental health impacts, as well as substantial social and economic costs [1–3]. Intimate partner violence (IPV) is defined by the World Health Organization as 'behaviour by a current or former intimate partner that causes physical, sexual and psychological harm, including acts of physical aggression, sexual coercion, psychological abuse and controlling behaviours' [4]. Women from refugee and immigrant backgrounds who are settling in a new country are at greater risk of IPV than women born in the country of settlement [5, 6].

Countries of resettlement have developed policies and programs to provide information, support and various forms of assistance for refugees [7]. In 2020, Australia ranked third, behind the United States and Canada, for resettling refugees referred by the Office of the United Nations High Commission for Refugees [7] and has built a network of services for migrants and humanitarian entrants that operate alongside mainstream services [8, 9].

While prevalence estimates are limited and vary in different jurisdictions [10], one third of refugee and migrant women in Australia are estimated to experience domestic and family violence [11]. However, refugee women are less likely to report IPV to police or use formal services and are more likely to remain in abusive relationships than locally-born women [12–15].

Vulnerability to IPV for refugee women during settlement is exacerbated by separation from family, exposure to pre-arrival trauma, limited social support networks [16, 17] and lower utilisation of mainstream health services [16, 18, 19]. Language, lack of knowledge about services and institutions, procedural hurdles, and visa status create additional barriers to help-seeking [20, 21]. At the same time, refugee and migrant women act with resourcefulness and agency during settlement [22], with many who are experiencing IPV exercising choice and agency, drawing on individual, family and community strengths to promote change [23].

Universal screening for IPV in health care services is recommended for priority populations [24] and involves asking all women a small number of standardised and validated questions about experiences of violence at home. Asking women directly about experience of violence, raises awareness about the issue, affirms women's right to safety, increases disclosure and creates opportunity for supportive intervention [25, 26]. IPV screening using validated tools has been implemented in diverse health settings including ante-natal clinics, primary health care, emergency departments, well baby clinics, substance treatment programs and mental health services [27, 28].

More than ten validated IPV screening tools have been utilised in healthcare settings, most frequently the HITS tool [29]. However, examination and practice feedback have identified limitations with the tools including the length, scoring variability and failure to address fear and coercive control behaviours [30]. Consequently, a new IPV screening tool was developed by Hegarty and colleagues [30] and validated against the Composite Abuse Scale [31, 32]. The four-question tool, titled ACTS, asks women how often in the last 12 months a partner or former partner has made them **A**fraid, **C**ontrolled, **T**hreatened or **S**lapped/ physically hurt them, with responses scored on a frequency scale with options for Never (0), Rarely (1), Sometimes (2), Frequently (3) and Very frequently (4). A positive screen for IPV ranges from the threshold score of 1 to a maximum of 16 and provides an indication of the frequency of exposure to IPV Evaluation of the ACTS tool found high levels of sensitivity and specificity in correctly identifying women with and without IPV when validated against the referent Composite Abuse Scale [30]. Further testing of this new tool in practice settings is warranted.

Being asked about IPV during routine visits to a health provider, combined with follow up support, has been found to have moderate net benefit and a low risk of adverse outcomes for

women [24, 33]. These interventions are acceptable to the great majority of women if screening questions are asked in a sensitive way within a safe environment [25, 28, 33, 34].

Within health settings, barriers to IPV screening with women from culturally and linguistically diverse background have been identified, including language barriers, problems with the use of interpreters, lack of knowledge of available services, limited understanding of what constitutes family violence in the country of settlement, and insecure visa status [21, 35, 36]. However, screening of refugee women in settlement services is untested and refugee women's views about the acceptability of IPV screening are unknown. Settlement services are frequently accessed by refugee women and have the potential to be suitable sites for IPV identification and response.

The study reported here explored refugee women's experience of IPV screening while accessing settlement services after arrival in Australia, their views about the acceptability of the screening questions and factors that enable and/or inhibit disclosure. This was part of a larger research project (publications pending) investigating culturally sensitive IPV screening and response for refugee women using a mixed methods approach that included interviews with refugee women, focus group discussions with settlement staff, interviews with service managers, screening and response data.

## Methods

### Study sites

The Australian Government operates a permanent settlement program for refugees with the number of places determined annually. In 2016–17, the program had 13,750 places, increasing to 16,250 places in 2017–18 and 18,750 places in 2018–19 [9]. Support for refugees is provided by government funded services delivered by community organisations across Australia, mainly in the major urban centres of Sydney and Melbourne. The study was conducted with women accessing one of the principal support services, the Settlement Engagement and Transition Support service (SETS)that provides support for refugees from about 18 months to five years post-arrival [37]. SETS is a follow up program to the intensive case-management support service provided to refugees for the first 18 months after arrival in Australia, the Humanitarian Support Program (HSP) [38].

Families and individuals are referred to a SETS provider with the referral including visa and other relevant information that confirms their eligibility to access the SETS service. Using the SETS service is voluntary. SETS provide a wide range of services including introduction to the legal, education and health systems; language courses; citizenship courses; assistance to obtain short- and long-term accommodation; access to employment; transport; civic participation; personal and family issues [39, 40].

Four Australian SETS services were selected as study sites, three in metropolitan Sydney and one in a major NSW regional centre. SETS service sites were chosen on the basis of having large and diverse client populations, readiness to participate in the study, and locational accessibility to the research team.

### Intervention

The intervention comprised screening by caseworkers and referral to an onsite IPV specialist worker for risk assessment, safety planning and further referral as needed. In preparation, SETS caseworkers and IPV specialist workers at the four sites received training from research team members, who were experienced IPV trainers, on the nature and impacts of IPV, barriers to disclosure, context for refugee women, the screening tool, responding to disclosure and study processes. The two-day training program was delivered face to face at each service site. IPV specialist workers from all sites participated in an additional half day of training on

response tools. During the four-month study period members of the research team met regularly, every three weeks on average, with participating staff to provide support and debriefing opportunities, enable staff to raise questions about the intervention, build capacity and help ensure fidelity of the research.

The trained caseworkers administered the ACTS IPV screening tool and offered all women a discreet wallet-sized IPV information card produced by the NSW Ministry of Health. The ACTS tool was translated into the four languages most spoken by women visiting the sites (Arabic, Farsi, Chinese and Vietnamese) and the information cards provided to the sites by the research team were in the four main languages as well as English.

Women identified as experiencing IPV were provided with a first-line response centred on listening and validation in accordance with the WHO LIVES model [41] and an offer of internal referral to the service's designated IPV worker. The IPV workers were given additional training and tools based on DOVE (Domestic Violence Enhanced Perinatal Nurse Home Visiting Program), an evidence-based brief psycho-educational intervention from the USA [27, 42, 43]. Intervention discussions were guided by an eight page booklet adapted from the DOVE resource [42, 43] with consent of its authors and in consultation with a panel of refugee women. The intervention included discussion of IPV impacts, options, safety planning, and risk assessment using the Danger Assessment for Immigrant Women (DA-I) [44].

## Follow up survey design

A follow up survey was administered three months after the intervention and comprised multiple pathways with skip logic built in to allow tailored questions for three participant groups: (i) those that did not remember or were not asked the screening questions, (ii) those who remembered the screening and told the caseworker at the site that no IPV was occurring, and (iii) those who disclosed IPV to the caseworker in response to the IPV screening. Participants in groups (i) and (ii) were asked the ACTS screening questions as part of the follow up survey. Other items included non-identifying information about participants' backgrounds, household composition, experience of using the SETS service, recollection of IPV screening, acceptability of the screening questions, experience of IPV (if applicable), services received and steps toward safety. Additionally, the Kessler Psychological Distress Scale (K6) [45], post-migration difficulties questions [46] and Composite Abuse Scale (Revised) Short Form (CAS-SF) [47, 48] were embedded in the survey (results to be reported separately).

Regarding acceptability, women who recalled being asked the screening questions responded to an item about their level of 'comfort' with the screening questions. All participants also responded to an item about their level of 'agreement' with settlement services asking questions of women about experiences of being frightened, controlled, or hurt. Participants were also invited to provide open text responses to supplement their response. This approach recognises important differences between comfort and agreement noting that women may feel uncomfortable with the personal and sensitive nature of the questions but nevertheless agree it is a useful or necessary process [49]. Survey participants were asked an open-ended question regarding the reasons for their agreement or disagreement with settlement services asking women about being frightened, controlled or hurt. In addition, they were asked, 'if settlement services staff are going to talk to women about this issue, what sort of things are important or might make a difference to women?'.

## Participant recruitment

Inclusion criteria were: i) female; ii) aged 18 years and over; and iii) accessing a SETS program. Women visiting the study sites for any reason during the four-month period, 1 March– 30

June 2022, were approached on arrival at the service by a member of the study's recruitment team who had undergone prior training from the research team. Women were provided with written and verbal information about the study and invited to consent for re-contact in three months to participate in a telephone survey about women's health and safety, and their experience of the service. Participant information and consent forms were translated into the predominant community languages (Arabic, Farsi, Chinese and Vietnamese) as well as English. Following provision of the written and verbal information, written consent was obtained. The initial consent recorded the participant's preferred language for the telephone survey, contact number, preferred time to call and times not to call.

## Survey implementation

The survey was administered three-months after recruitment by a team of bilingual research assistants (RAs), sourced from a pool of bicultural/bilingual guides employed by study partner SSI. All had experience supporting refugee women and prior experience conducting telephone surveys. Most were former refugees and had first language proficiency in the main languages spoken by participants: Arabic, Farsi, Dari, Chaldean, Assyrian, Mandarin, and Vietnamese. Participants needing other languages were surveyed with the support of the national fee-based telephone interpreter service.

The RAs received two days training covering the impacts of IPV, context for refugee women, telephone interviewing skills, consent procedures, the study's safety and privacy protocol, survey questions, data recording and other study processes, with opportunities to practise the survey. Survey calls were made from a dedicated space in an office of the partner organisation, SSI, with 1–5 RAs rostered 2–3 days per week, making calls on the days and times nominated by participants during the first stage of recruitment. Onsite support and supervision were provided by members of the research team.

When participants were contacted, they were provided further information about the survey, specifically that some of the survey questions related to IPV and the IPV screening recently introduced at the study site. Verbal consent to proceed with the survey was then sought from participants by the RAs and recorded on the survey assent form. The study's safety and privacy protocol included checking with the participant that her situation was safe, private, and convenient, and letting her know that she could stop answering questions at any time, before commencing with the survey questions.

The follow-up period of three months was selected as a time period found in other research to be sufficient for possible action by women following the intervention, and soon enough for most women to remember the intervention [50, 51], while recognising that women choose many different pathways and time frames for action [52, 53] Surveys were conducted by phone in the participant's nominated language using the survey tool designed by the research team and implemented with Qualtrics XM survey software [54]. No identifying information was requested or recorded during the survey. The average call time was 28 minutes. Women who participated in the survey were offered a $35 shopping voucher to acknowledge the value of their time. Survey data was recorded in Qualtrics and exported to Excel, csv and SPSS v29.

## Data analysis

Descriptive data is presented for the sample demographics and frequencies for participants' acceptability (comprising comfort and agreement with screening). Cross tabulations were carried out for screening comfort and agreement, with IPV disclosure during the visit or the survey. Additionally, cross tabulations were carried out for screening comfort and agreement with age, country of birth, language spoken at home, time in Australia, household composition

and number of visits to the SETS service. Fisher's exact test was used to determine associations between variables. Content analysis was conducted to categorise open-ended question responses. For the open-ended questions regarding 'what sort of things are important or might make a difference to women' in screening women for IPV within settlement services, unprompted responses were coded by Research Assistants at the time against a prepared list of eight enablers from the literature: care shown by the worker, talking to someone in my own language, trust in the confidentiality/privacy of the service, talking to a female worker, knowing and trusting the worker, feeling confident the worker will know what to do, talking to a worker who has a similar background to me, and talking to a worker who has a different background to me. If women offered no response, the list of factors was read and prompted responses were recorded.

## Ethics

The project was guided by a panel of former refugee women with lived experience and an advisory group comprising representatives from key IPV and refugee peak agencies. The WHO ethical and safety guidance on conducting research with women who have experienced violence was followed [55]. For example, screening with women was not conducted in the presence of any persons aged 3 years or over. No identifying information was recorded on the screening forms to protect women's privacy and safety. Approval was granted by the University of Wollongong Human Research Ethics Committee (2021/388). Written consent was obtained during site visits, 1 March 2022–30 June 2022, and verbal consent for the survey was obtained and recorded by RAs during the survey period 17 July 2022–7 October 2022.

## Results

### Participants

In the four-month period between March and July 2022, 429 women visiting the service sites were invited to be contacted for the survey of whom 375 (87%) gave consent. Reasons for declining consent were not recorded. Of the 375 women who consented to follow up contact, 321 were able to be contacted by the bilingual RAs three months after the service visit, and consented to participate in the survey, a response rate of 86%. Another 32 women were unable to be contacted, 19 declined and three surveys were not completed.

It should be noted that the screening group and survey group do not exactly align. Some women recruited in the waiting area of the site did not subsequently meet with a caseworker and as a result, were not asked the screening questions. It is also the case that some women were asked the screening questions on days when the study recruitment team were not present and, as a result, were not included in the survey.

### Participant profile

Key demographic data for survey participants are in Table 1 with other characteristics including: 25 languages were spoken at home with the most spoken language being Arabic (40%, n = 127), followed by Chaldean (11%, n = 36), Assyrian (9%, n = 30), Dari (9%, n = 30), and Mandarin (8%, n = 25). Most women (67%, n = 216) had various Class 200 visas designating them as 'refugees and humanitarian entrants' while 12% (n = 39) were unsure of or did not state their visa type. The number of visits to the SETS service reported by participants ranged from once only (37%, n = 118) to more than 10 visits (20%, n = 65) with the types of assistance most frequently sought being: information, advice or referral (75%, n = 239), followed by group activities or events (36%, n = 115), education/training/employment (32%, n = 101), and

**Table 1. Participant characteristics (n = 321).**

| Age (mean = 44.3years) | n (%) |
|---|---|
| 18–25 | 22 (7) |
| 26–35 | 75 (23) |
| 36–45 | 77 (24) |
| 46–55 | 79 (25) |
| 56–65 | 43 (13) |
| ≥ 66 | 24 (8) |
| Not stated | 1 (0) |
| **Household composition** | |
| Husband/partner and children +/- others | 184 (57) |
| Husband/partner | 38 (12) |
| Husband/partner +/- others | 7 (2) |
| Children +/- others | 33 (10) |
| Others (excluding husband/partner and children | 35 (11) |
| Alone | 24 (8) |
| **Country of birth** | |
| Iraq | 151 (47) |
| Syria | 43 (13) |
| China | 35 (11) |
| Afghanistan | 32 (10) |
| Iran | 11 (3) |
| Vietnam | 10 (3) |
| Congo | 5 (2) |
| Myanmar | 5 (2) |
| Sri Lanka | 5 (2) |
| Eritrea | 4 (1) |
| Ethiopia | 4 (1) |
| Others (Egypt, Pakistan, Burundi, Hong Kong, Jordan, Lebanon, Morocco, Palestine, Sudan, Taiwan, Tanzania, Tunisia, Turkey) | 16 (5) |
| **Time since arrival in Australia** | |
| More than 5 years | 45 (14) |
| 3–5 years | 221 (69) |
| 1–2 years | 46 (14) |
| Less than 1 year | 9 (3) |
| **TOTAL** | **321 (100)** |

financial advice/income support/money matters (26%, n = 82). (Participants accessed multiple forms of assistance so percentages total >100%.)

Participants said what they most liked about the SETS services were the staff being caring/ helpful/ supportive (87%, n = 279), followed by talking to people in my language (68%, n = 217), the information provided by the service (65%, n = 209), and the service being close to where I live or easy to access (49%, n = 157). (Participants identified multiple reasons for liking the service so percentages total >100%).

## Screening status

Fifty six percent (n = 180) of the survey participants recalled being asked the screening questions at the SETS services. This was lower than anticipated. Likely reasons for the lower rate,

**Table 2. Participants identifying IPV at service visit or during follow up survey (n = 316[a]).**

| | | Follow Up Survey IPV Screening | |
|---|---|---|---|
| | | Frequency (% of total) | |
| | | Positive | Negative |
| Visit IPV Screen | Positive | 27 (9) [b] | - |
| | Negative | 8 (3) | 145 (46) |
| | Don't Know[c] | 13 (4) | 123 (39) |
| | **Total** | **48 (15)** | **268 (85)** |

Table notes: a = five missing responses; b = Those who screened positive for IPV during their visit were not asked the IPV screening questions during the survey; c = No recall of screening or not asked at their visit.

based on consultation with caseworkers at the sites were: inclusion in the survey cohort of some participants who were not screened, participants managing multiple pressures and simply not recalling the questions, and memory impacts of the three-month follow-up interval.

During the survey, participants who did not recall the screening questions and participants who indicated they had not disclosed when visiting the service, were asked the ACTS questions again under survey conditions of anonymity. Eight women who said they had not disclosed at the service and another 13 women who had said they weren't asked or didn't recall being asked, screened positive for IPV during the survey. This gave a total of 48 women who identified IPV, either during the visit or in survey responses (Table 2). Five participants in the group who said they weren't asked or didn't recall being asked at the service and did not answer the screening questions during the survey, were excluded from the analyses.

## First or prior experience of IPV screening

Of the 27 women who indicated they had disclosed IPV at the SETS service in response to screening, 13 (48.1%) said that was the first time they had disclosed IPV. Women who did not recall screening and women who screened negative at the service visit (n = 289) were asked about being asked IPV questions by other services. Only 41 women in these categories (14%) reported being asked questions about IPV at other services (Table 3). For most women the study intervention appears to be their first experience of IPV screening.

## Acceptability: Comfort with being asked about IPV

The 180 women who remembered being screened at the services were asked during the survey how comfortable they were with being asked about being frightened, controlled, or hurt by their partner. Eighty two percent of participants were 'very comfortable', and eleven percent

**Table 3. IPV screening by other services (n = 289).**

| | Frequency (%) | | |
|---|---|---|---|
| **Remember screening by other services** | IPV screening status | | Total |
| | Positive | Negative | |
| Not screened by other services | 13 (62) | 235 (88) | 248 (86) |
| Screened by other services | 8 (38) | 33 (12) | 41 (14) |
| Total | 21 (100) | 268 (100) | 289 (100) |

Table excludes the 27 women who disclosed during the visit and the five participants who were missing in the response to the screening questions during the survey. Fisher's exact p = 0.004.

**Table 4. Comfort with IPV screening at visit (n = 180).**

| How comfortable/ uncomfortable were you in being asked these questions? | Frequency (%) | | Total |
|---|---|---|---|
| | IPV screening status | | |
| | Positive | Negative | |
| Very comfortable (fine to be asked / no problem) | 23 (66) | 125 (86) | 148 (82) |
| Reasonably comfortable | 8 (23) | 12 (8) | 20 (11) |
| A bit uncomfortable (unsure/not sure it was appropriate) | 3 (9) | 5 (3) | 8 (4) |
| Very uncomfortable (offended/upset) | 1 (3) | 3 (2) | 4 (2) |
| Total | 35 (100) | 145 (100) | 180 (100) |

The no recall of screening group were not asked this question (n = 141). Fisher's exact p = 0.02

were 'reasonably comfortable' with being asked the ACTS screening questions (Table 4). A significant difference in the degree of comfort was found between women who disclosed IPV and those who didn't, with 86% of women who did not disclose being 'very comfortable' compared to 66% of the women with direct experience of IPV (Fisher's exact p = 0.02).

## Acceptability: Agreement with being asked about IPV

All survey participants, including those who weren't screened or didn't remember being screened, were asked if they agreed with settlement services asking women about being frightened, controlled, or hurt. High levels of agreement were found with more than 84% of the total sample 'strongly' or 'somewhat' in agreement, while 14% were strongly or somewhat in disagreement. There was no significant difference in agreement levels between women who did (87%) and didn't (86%) screen positive for IPV (Fisher's exact p = 0.99) (Table 5).

Differences were found in the level of agreement between those who weren't asked or didn't remember being asked the IPV screening questions, and those who remembered being asked the questions. Those with no recall of the questions had lower levels of overall agreement (77%) and higher levels of disagreement (19%) compared to those who recalled being asked the screening questions (88% and 10% respectively).

## Factors associated with acceptability

No statistically significant differences in either comfort or agreement with IPV screening were associated with age, country of birth, household composition, time since arrival in Australia, visa type or number of visits to SETS. In relation to language spoken at home there was no difference in comfort with IPV screening, but some differences were found for agreement with

**Table 5. Agreement / disagreement with screening (n = 316).**

| Do you agree that services like [NAME OF SITE] should ask women about being frightened, controlled or hurt? | Frequency (%) | | Total |
|---|---|---|---|
| | IPV screening status | | |
| | Positive | Negative | |
| Strongly or somewhat agree | 41 (85) | 223 (83) | 264 (84) |
| Neither agree nor disagree | 1 (2) | 7 (3) | 8 (3) |
| Strongly or somewhat disagree | 6 (13) | 38 (14) | 44 (14) |
| Total | 48 (100) | 268 (100) | 316 (100) |

Fisher's exact p = 0.99

IPV screening. Specifically, women who spoke Arabic were more likely to strongly or somewhat agree, 88% (111/126), compared to 83% for the total sample and, while noting that the numbers are small, 28% (10/36) of women who spoke Chaldean and 27% (8/30) of women who spoke Assyrian somewhat or strongly disagreed that services should ask IPV questions compared to 14% of the total sample (Fisher's exact p = 0.03).

## Narrative responses

Following the questions about agreement or disagreement with settlement services asking women about being frightened, controlled or hurt, all participants were asked to provide an explanator comment. Of the 321 survey participants, 314 (98%) provided a comment. The research team coded comments to one of a list of reasons proposed by the research team, as reported in Table 6.

The most common (159/ 314) reasons for agreement with IPV screening related to it being a means for women getting help or being kept safe or protected. Women who had disclosed IPV commented, for example, that they agreed with screening, *To get help like I did (D19)*, and *Because many new arrivals, women may be subject to DV verbal abuse, physical abuse, etc and they need support to navigate and seek help (NA46)*. Other comments (51/314) related to women's empowerment and/or feeling unburdened by speaking about IPV, such as these quotes from women who disclosed IPV:

*When you get to talk to the right person, speak out about what happened, I feel lightened and find out solutions to my problems (D11).*

*There is someone who hears me without judging me and this helps to release the internal pressure (D21).*

*I know my rights (D26).*

Women also commented that asking the questions could help to overcome barriers to talk about DV, such as, *Many women [are] afraid, hesitate to talk about DV, but when having a*

**Table 6. Reasons for agreement/disagreement with settlement services asking women about being frightened, controlled or hurt (n = 314).**

| | IPV Screening Status | | |
|---|---|---|---|
| | Positive | Negative | Total |
| **Reasons for agreement** | | | |
| 1. So women can get help/ be kept safe | 20 | 139 | 159 |
| 2. To enable women to talk about IPV | 7 | 38 | 45 |
| 3. To increase awareness/education about IPV | 3 | 8 | 11 |
| 4. To empower women; unburden women; realise women's rights | 13 | 38 | 51 |
| 5. Other | 1 | 18 | 19 |
| **Reasons for disagreement** | | | |
| 6. Discomfort | 0 | 4 | 4 |
| 7. Opposed | 0 | 3 | 3 |
| 8. Privacy/confidentiality concerns | 1 | 2 | 3 |
| 9. Questions unnecessary | 0 | 2 | 2 |
| 10. Other | 0 | 0 | 0 |
| **Reason unclear (could be agree or disagree)** | | | |
| 11. Reason not clear | 3 | 14 | 17 |
| **Total** | **48** | **266** | **314** |

*chance to talk about it is a good thing (ND48)*, and, *Because many people experience DV does not have enough courage to talk about it unless someone ask about it (D14)*. A woman who did not report IPV agreed with screening *Because some women are subjected to violence and are unable to disclose and speak, but these questions help women to say their feeling (NA50)*. The value of IPV screening for raising awareness was also noted, *I think we can speak out, sharing the experiences we have been through, other people can understand more about DV and from different perspective (D6)*.

Twelve participant comments expressed opposition to the questions being asked, or queried their necessity. As one woman who had experienced IPV noted, *The worker ha[s] to be 100% make sure all these things are confidential because of our culture connection (D15)*. Other comments from women who had not disclosed abuse included:

> *I have no issues with my husband or my kids and I do not like those kind of questions (NA86).*

> *Family secret must keep secret and should not tell others about it (NA15).*

> *Women shouldn't tell others family problems because they might advise her to divorce (NA32).*

## What would make a difference

In response to the item on conditions that would make a difference for women in talking about IPV with settlement services staff, unprompted responses were coded by Research Assistants against a prepared list of eight factors or, if women offered no response, the list was read and prompted responses were recorded. The most highly ranked factors were 'care shown by the worker', 'talking to someone in my own language', 'trust in the confidentiality/privacy of the service' and 'talking to a female worker' (Table 7). These four factors were ranked highly by all women–those who disclosed IPV and those who didn't. The women who disclosed IPV also identified 'knowing and trusting the worker', and 'feeling confident the worker will know what to do' as important enabling factors (Table 7).

## Discussion

This study explored refugee women's experience of an intervention to identify and respond to IPV introduced in Australian settlement services. When surveyed three months after the

**Table 7. Factors that would make a difference for women talking about IPV (n = 316).**

| Factor that would make a difference | Frequency (%) | | |
| --- | --- | --- | --- |
| | IPV Screening Status | | |
| | Positive | Negative | Total |
| Care shown by the worker | 36 (75) | 192 (72) | 228 (72) |
| Talking to someone in my own language | 35 (73) | 197 (74) | 232 (73) |
| Trust in the confidentiality/privacy of the service | 33 (69) | 156 (58) | 189 (60) |
| Talking to a female worker | 25 (52) | 163 (61) | 188 (59) |
| Knowing and trusting the worker | 24 (50) | 123 (46) | 147 (47) |
| Feeling confident the worker will know what to do | 26 (54) | 87 (32) | 113 (36) |
| Talking to a worker who has a similar background to me | 16 (33) | 75 (28) | 91 (29) |
| Talking to a worker who has a different background to me | 1 (2) | 9 (3) | 10 (3) |
| Other* | 0 (0) | 8 (3) | 8 (3) |
| **Total** | **48 (100)** | **268 (100)** | **316 (100)** |

\* Other included: Making the woman feel safe (3) and the caseworker not being biased towards women (1). Multiple responses were allowed so column totals are>100%

intervention, women indicated high levels of comfort with being asked questions about being afraid, controlled or hurt by their partner, and expressed agreement with settlement services routinely asking such questions. The strong acceptance of IPV screening among refugee women found in our study is consistent with the growing evidence indicating that women endorse direct asking about experiences of violence in order to encourage disclosure and create opportunities to receive help [25, 33, 56]. However, most of the research to date has been in healthcare settings whereas this study appears to be the first to investigate refugee women's experience of IPV interventions conducted while accessing settlement support services.

Significantly, women experiencing IPV had lower levels of comfort while, at the same time, expressing strong agreement about settlement services asking these questions. This suggests that for women experiencing IPV, talking about their experience is difficult, but they believe it is important for settlement service providers to ask in order to give women the chance to talk. It is also noteworthy that women who weren't asked the questions or did not remember being asked, had lower agreement and higher disagreement than those who remembered being asked the questions, indicating that women are inclined to be more in favour when they have had direct experience of being asked. While the great majority of participants were in agreement with being asked the screening questions, 14% were not in favour with responses indicating concerns about intrusion into family privacy, discomfort with the questions, concern about confidentiality and believing the questions were not necessary. It suggests that additional explanation may be needed from settlement service caseworkers to ensure women are informed about the context and rationale for screening, and have opportunity to discuss any concerns.

For most women in the study, the intervention was their first experience of being asked about IPV. This suggests that the opportunities for refugee women to disclose in response to direct asking by service providers are limited. Women's responses in the study also point to the benefits of screening that extend beyond disclosure of IPV, with participants commenting on the value of creating opportunities for women to get help, education and awareness raising about IPV, affirming women's rights to safety and empowering women who may feel isolated. Care shown by the worker has been highlighted by other studies as one of the most important factors to enable women's disclosure [25, 57]. This is confirmed by our study as being of high importance to refugee women, with those experiencing IPV ranking it as the most important factor. Spangaro, Koziol-McLain et al. (2016) identified 'direct asking' and 'care shown by the worker' as the two most important conditions for decisions to disclose for women accessing antenatal care, with 'care' based on perceptions of worker trustworthiness, showing interest and a non-judgemental attitude [57].

Also consistent with other research is refugee women's preference, identified in our study, for speaking with a female worker [58], as well as the importance of perceiving that the discussion will be treated confidentially [25] and having confidence that the worker will know what to do if the woman chooses to disclose [25, 57]. These are all confirmed as important conditions for refugee women to encourage discussion of IPV.

Our study finds that refugee women place a particularly high value on being able to talk with someone who speaks the same language. This was the highest ranked enabler for the full sample and the second highest for women who disclosed IPV. Our research points to language matching and culturally safe service environments as important enablers for refugee women to decide to disclose and seek help. In this regard, the service setting offered by settlement organisations emerges as a key finding in our study. Refugee women value the proximity, accessibility, care shown by staff, cultural safety, and ease of being able to communicate in their language. The inclusive, community-based setting serves to promote well-being [59] and is a conducive environment for conversations about a range of complex issues such as responding

to IPV. At the same time multiple barriers to speaking up about IPV for refugee women are identified (reported separately) including fear of retribution, concerns about the consequences of disclosure, not wanting to break up the family, economic insecurity, lack of knowledge about Australian laws and services, visa insecurity, and complex relationships with communities [60].

Knowing and trusting the worker is identified in our study as important which suggests that building rapport is an important condition for refugee women. However, it is also the case that many women in our study were visiting the settlement service for the first time and no association was found between the number of visits to the settlement service and disclosure, which suggests that rapport can be quickly established by skilled, caring workers.

## Limitations

It is notable that the study design did not include a comparison group for testing the effectiveness of the IPV screening and response. This reflected a change in the study design that had been required due to COVID-19 and at a time when there was almost no refugee intake in Australia. It is also the case that some survey participants in the survey may not have received the IPV intervention due to caseworkers not carrying out the intervention. These participants' views are not based on direct experience of the intervention but, nonetheless, are responses to the survey questions and based on their experience of interacting with the settlement service.

## Implications

This study demonstrates that screening for intimate partner violence is acceptable to recently arrived refugee women in the context of refugee settlement services. This is an important finding for this population, which is often hard to reach. Policy and programming should consider building IPV identification and response as one of the deliverables in refugee settlement service provision. This study was undertaken in the SETS stream which responds to women 18 months to five years post-arrival. Further testing in the immediate on-arrival service stream is also indicated, given the different stressors women face in the earlier arrival period. The successful implementation of the intervention in this study was assisted by the active role of the research team in providing pre-intervention training, regular debriefing opportunities with implementing staff, refresher training and study tools. Training, ongoing support, organisational mandate, referral protocols and user-friendly tools in community languages are required for effective implementation of IPV screening and response in settlement services.

## Conclusion

Direct asking about IPV is found in this study to be acceptable to refugee women accessing settlement services. Settlement services provide a setting and an opportunity to identify and respond to IPV experienced by refugee women due to the capacity offered by caring workers, language matching, and sites perceived as safe and welcoming. Findings encourage consideration of further implementation in settlement services of a protocol involving the ACTS IPV screening tool followed by a listening and validation response, and referral to an internal IPV specialist.

## Author Contributions

**Conceptualization:** Jo Spangaro, Jacqui Cameron, Kelsey Hegarty, Jane Koziol-McLain, Tadgh McMahon, Chye Toole-Anstey, Anthony Zwi.

**Data curation:** Nigel Spence, Jo Spangaro, Nicola Man, Jacqui Cameron, Chye Toole-Anstey, Jeannette Walsh.

**Formal analysis:** Nigel Spence, Jo Spangaro, Nicola Man, Jacqui Cameron, Kelsey Hegarty, Jane Koziol-McLain, Tadgh McMahon, Astrid Perry-Indermaur, Chye Toole-Anstey, Jeannette Walsh, Anthony Zwi.

**Funding acquisition:** Jo Spangaro.

**Investigation:** Nigel Spence, Jo Spangaro, Jacqui Cameron, Chye Toole-Anstey, Jeannette Walsh.

**Methodology:** Nigel Spence, Jo Spangaro, Jacqui Cameron, Kelsey Hegarty, Jane Koziol-McLain, Tadgh McMahon, Astrid Perry-Indermaur, Chye Toole-Anstey, Anthony Zwi.

**Project administration:** Nigel Spence, Jo Spangaro, Astrid Perry-Indermaur.

**Resources:** Jo Spangaro.

**Supervision:** Jo Spangaro.

**Validation:** Jo Spangaro, Nicola Man, Jacqui Cameron, Kelsey Hegarty, Jane Koziol-McLain, Tadgh McMahon, Astrid Perry-Indermaur, Anthony Zwi.

**Writing – original draft:** Nigel Spence.

**Writing – review & editing:** Nigel Spence, Jo Spangaro, Nicola Man, Jacqui Cameron, Kelsey Hegarty, Jane Koziol-McLain, Tadgh McMahon, Astrid Perry-Indermaur, Chye Toole-Anstey, Jeannette Walsh, Anthony Zwi.

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
