## [Decision Letter · Decision Letter 0]

18 Sep 2024

PONE-D-24-04846The acceptability of intimate partner violence screening and response among refugee women accessing Australian resettlement servicesPLOS ONE

Dear Dr. Spence,

Thank you for submitting your manuscript to PLOS ONE. After careful consideration, we feel that it has merit but does not fully meet PLOS ONE’s publication criteria as it currently stands. Therefore, we invite you to submit a revised version of the manuscript that addresses the points raised during the review process.

Thank you for this important submission. As you will see from the reviewer comments, there is a great deal of support for this manuscript with some minor revisions. Please address the reviewer suggestions and we look forward to receiving your revised manuscript.

We look forward to receiving your revised manuscript.

Kind regards,

Michelle L. Munro-Kramer, PhD, CNM, FNP-BC, FAAN

Academic Editor

PLOS ONE

2. Please provide additional information regarding the considerations  made for the refugees included in this study. For instance, please discuss whether participants were able to opt out of the study and whether individuals who did not participate receive the same treatment offered to participants.

“This study was funded by the Australian Research Council (LP190101183) and SSI, a large non-government agency providing settlement services based in Australia.”

Reviewers' comments:

Reviewer's Responses to Questions

**Comments to the Author**

1. Is the manuscript technically sound, and do the data support the conclusions?

Reviewer #1: Yes

Reviewer #2: Yes

2. Has the statistical analysis been performed appropriately and rigorously? 

Reviewer #1: Yes

Reviewer #2: Yes

3. Have the authors made all data underlying the findings in their manuscript fully available?

Reviewer #1: No

Reviewer #2: Yes

4. Is the manuscript presented in an intelligible fashion and written in standard English?

Reviewer #1: Yes

Reviewer #2: Yes

5. Review Comments to the Author

Reviewer #1: The paper investigates acceptability of surveying refugee women in Australia about intimate partner violence (IPV). The sample includes refugee women visiting resettlement support centers who were asked about their acceptance of IPV screening by the resettlement services. The results indicate that women refugees are supportive of IPV screening because it may enable them to get help, empower them, enable talking about IPV and increase awareness of IPV. There is no differentiation by women’s origin, age and other demographic characteristics, which potentially indicates a broad appeal of the intervention. There are some issues that could be addressed to improve the paper.

1/ Some contextual information would be beneficial. For those who do not know Australian context, it would be good to provide more details about what the resettlement services do. In addition, the total number of resettlement service centers in the country and major cities so that we can judge the relative representativeness of the sample. How many visitors does a center receive per day or per year? How many (or what proportion) are refugee women?

2/ When explaining the ACTS tool on page 5, it is not clear how the scores are obtained. Are there four questions with four possible answers or more questions within each ACTS category? If four questions, does each question get a response 1-4 (or 0-4) and do these reflect severity or frequency of exposure to IPV?

3/ The stated rationale behind the study seems to be that refugee women’s views and acceptability of IPV screening in resettlement service centers are not known. It is not clear what the expectations are from theory or prior evidence for women feeling differently about screening at resettlement service centers as opposed to health care centers.

4/ It is mentioned that IPV screening is typically done in relation to health care providers visits. There are some downsides to it, but it is not very clear what the advantages of IPV screening at the resettlement service centers would be. I am not sure what the advantage is apart from resolving language barriers. This should come out clearer. Please elaborate. IPV screening in resettlement service centers could also have downsides. One potential downside could be that women might be afraid about talking about IPV at the same place they go to look for job (or seek information about jobs; this is not clear to me as I do not know the Australian context and the details are not given in the paper). If they fear that a future employer might discover something private about them such as IPV they may be reluctant to talk about it in resettlement centers. Similarly, they may fear visa or other paperwork being denied. I am not sure about how IPV is treated legally in Australia. Are there fines or sentences for perpetrators? If so, women may not be comfortable about IPV screening at resettlement service sites as they fear legal persecution or deportation for their partners.

5/ Were the shopping vouchers for a specific shop? If one particular shop (or a chain of shops), why was it chosen? Could the shop choice deter participation?

6/ It would be very useful to state in the methods section which eight enablers were used to code the qualitative responses.

7/ It is difficult to understand the numbers on participation. It is stated that out of 375 women that agreed to take part, 321 were surveyed, 30 were unable to be contacted, 19 declined and 1 survey was not completed. The sum of 321+30+19+1 is 371. What about the remaining 4? Also, for each discrepancy between the recruited and surveyed women, please give a number (e.g. excluded and not asked the screening questions).

Minor

More precise sentence phrasing would be beneficial. For example, the last sentence in the first paragraph would benefit from clearly stating the comparison group for IPV affected refuge women (e.g. host or origin country population). It is also not clear how vulnerability of refugee women should be understood in the last paragraph on page 4 (is this vulnerability to IPV or broader vulnerability?).

On page 8, it is stated in the four main languages as well as English and three other community languages. Please check if there is duplication here as later on only four languages in total were mentioned.

Change the indicator of affiliation from 2, 4, 9 and 10 to 1 as it appears that this is the same institution.

Reviewer #2: This was a very well-written and timely study that will be extremely valuable globally! I really appreciate the care that the authors put into the study. To be honest, I found little to squabble with, but mostly have comments that I hope can strengthen what is already a strong piece of work. Thank you for doing this work and reporting on it.

Overall Comments:

There is an unstated assumption in this article that the primary purpose of screening women for IPV is to get a disclosure. But we know that survivors experience numerous barriers to disclosure and may not disclose when asked. Screening certainly provides survivors the opportunity to disclose their experiences and receive support and referrals to services. But it also may have benefits for survivors beyond disclosure, particularly when done in a trauma-informed and empowerment-centered way. While an older article, Spangaro, Zwi, and Poulos (2011)’s qualitative work suggests that being asked and naming abuse was important. More recently, work by Futures Without Violence recommends universal education and empowerment, and while that may not include asking a direct question, still suggests that initiating conversations can be beneficial. My primary suggestion in saying this is to be clear about the goals of screening women in refugee settings – is it simply to encourage disclosure? This is the outcome the authors most frequently discuss, at the exclusion of other benefits that can come from broaching the subject.

Specific Sections:

Introduction:

Line 71: “Universal screening for IPV in health care settings is recommended…”

Methods:

Line 152: rather than a comma after “groups” there should be a colon (:)

Participant Recruitment (Pg 9): were women who went to the center multiple times over that time period asked multiple times to participate in the survey? Also, the authors refer to “Chinese” as a language, but later it sounded like it was Mandarin? Maybe this could be more specific.

Pg 10 line 202: To make it clear that participants were only contacted once by RAs, maybe start this with “When participants were contacted, they were provided further information…”

Line 207: I found the phrase “before commencing the survey with the opportunity to stop the survey at any time” really awkward. Maybe “The study’s safety and privacy protocol included checking with the participant that her situation was safe, private, and convenient, and letting her know that she could stop answering questions at any time before commencing with the survey questions.”

Pg 11 Line 209: the follow up period was three months – I just wonder if the authors have any data describing how long it often takes women between a disclosure of violence to when they take some type of action?

Line 213: how were the participants who disclosed violence given the shopping voucher if they were living with the person causing harm?

Results:

Pg 12/13, Participant Profile: Did women have to show visas to receive services at SETS? In other words, was it clear if any of the participants were in the country w/o authorization or legal status? It would be good to know how this could affect screening and response.

Pg 16, First or prior experiences of screening: For the 13 women who said that the SETS screening was the first time they disclosed, were they asked if it was the first time they had been asked? Can the authors explain why they removed the 27 participants who said they screened positive for IPV during the initial visit from the analysis of being screened by other services? It seems like useful information to know if they had ever been screened elsewhere, regardless of whether they disclosed.

Pg 18, line 336: this might be better located in the discussion. Also, I think the authors mean that women who remember being asked are more in favour, not women who were asked.

Pg 19: there is a typo in either Table 6 or in the text where referring to the number of respondents say “So women can get help/be kept safe.” The table has 159, the text says 156/314.

Pg 20 lines 383-386: are these from different women or from the same person? The formatting makes it hard to tell. The authors could consider referring to individual respondents as “Participant 1” if they do not feel comfortable using additional descriptors.

Pg 20-21, What would make a difference: This section is confusing as it is written, particularly the part on

Pg 21 that tries to describe who rated which statements highly. I understood what the authors meant, but only after looking at the table closely.

Discussion:

This might be a place where my comments about the intention of screening always being disclosure could be addressed.

While I do not want to put undue weight on the responses of people who disapproved or would be uncomfortable with IPV screening, is it possible for the authors to provide a little interpretation about the reasons those few people gave against screening, and what that might suggest about the approach that refugee service providers take? I realize these were very few in number compared to those who support this practice, but providers often say that fear of offending patients (particularly related to cultural relativism) is one reason why they don’t want to talk about IPV, so acknowledging that while some participants felt uncomfortable with being asked, it was far from the majority. We need to move away from stereotypes about the overall acceptability of IPV among some communities.

Limitations: I do not understand how service improvement by SETS to better respond to clients experiencing IPV was a limitation of this study, looking at the acceptability of IPV screening in the setting. In terms of the research team offering assistance to the SETS sites, it would be useful to reflect on what that assistance was (presumably IPV training, support with asking difficult questions, etc.) and rather than seeing that as a limitation, reflect on it as a necessary part of a screening program for other refugee assistance settings.

Is an additional limitation that some participants in the survey did not actually receive the intervention, but cannot be separated from those who just don’t remember if they received the intervention? That seems worth reflecting on.

Tables

I will let the copy editors work with authors on formatting the tables. I don’t usually see shaded lines, and while I figured out what “IPV+ve” and “IPV-ve” mean, I don’t think it’s standard. It’s also nice when I see information about the population included in the title, even just something as simple as “all participants'

6. PLOS authors have the option to publish the peer review history of their article (what does this mean?). If published, this will include your full peer review and any attached files.

Reviewer #1: No

Reviewer #2: No

---

## [Author Response · Author response to Decision Letter 0]

18 Oct 2024

The 5 Journal Requirements and the 33 comments from the two reviewers have been very welcome in improving the manuscript. All points have been addressed in the eleven-page Response to Reviewers, attached, and the manuscript has been amended accordingly.

---

## [Editor Report · Decision Letter 1]

21 Nov 2024

The acceptability of intimate partner violence screening and response among refugee women accessing Australian resettlement services

PONE-D-24-04846R1

Dear Dr. Spence,

We’re pleased to inform you that your manuscript has been judged scientifically suitable for publication and will be formally accepted for publication once it meets all outstanding technical requirements.

Kind regards,

Michelle L. Munro-Kramer, PhD, CNM, FNP-BC, FAAN

Academic Editor

PLOS ONE

Additional Editor Comments (optional):

Thank you for the careful attention to addressing all reviewer comments. I look forward to seeing this manuscript in print.

---

## [Editor Report · Acceptance letter]

3 Dec 2024

PONE-D-24-04846R1 

PLOS ONE

Dear Dr. Spence, 

I'm pleased to inform you that your manuscript has been deemed suitable for publication in PLOS ONE. Congratulations! Your manuscript is now being handed over to our production team.

Kind regards, 

on behalf of

Dr. Michelle L. Munro-Kramer 

Academic Editor

PLOS ONE